# The Historic City, Its Transmission and Perception via Augmented Reality and Virtual Reality and the Use of the Past as a Resource for the Present: A New Era for Urban Cultural Heritage and Tourism?

**Diego A. Barrado-Timón** and **Carmen Hidalgo-Giralt** *

Department of Geography, Autonoma University of Madrid (UAM), c/Francisco Tomás y Valiente 1, 28049 Madrid, Spain; diego.barrado@uam.es

\* Correspondence: carmen.hidalgog@uam.es; Tel.: +34-914974587

**Abstract:** The objective of this study is to analyze the impact that augmented reality (AR) and virtual reality (VR) are having on our conception, appreciation, and use of urban heritage spaces. Although most evaluations that appear in the specialized literature are clearly positive in this respect, there is a critical line of thought that considers these new technologies as connected to prior theoretical assumptions about heritage, in terms of what we value, how we value it, and for what reasons. To contrast the two perspectives, we have selected and examined scientific literature evaluating the application of AR and VR in urban heritage spaces, in order to analyze whether, in addition to positive effects, certain negatives linked to the 'virtualization' of space are also at work. A qualitative methodology has been developed supported by the ATLAS.ti tool (Scientific Software Development GmbH, Berlin, Germany), which allows definition of the different thematic lines treated in the literature as well as the connections between them. Our main conclusion is that concerns around the critical aspects are very limited, with only a few perceiving the possible dangers of trivialization of heritage, the creation of virtual tourist worlds separate from the material space of socio-economic relations, negative effects on the way in which knowledge is constructed, or the difficulties for some user groups in accessing these technologies.

**Keywords:** augmented reality; virtual reality; cultural heritage; historical city; urban dynamics; ATLAS.ti

## 1. Introduction

Concepts of "heritage", "urban heritage", and "the historic city" are not universal and have not been constant in the societies where they arise. On the contrary, such terms first appeared in the West in the 19th century and were not definitively consolidated until the early decades of the 20th century. Since that time, they have assumed enormous significance in theoretical and practical works regarding the development of numerous cities.

A more recent addition to the complexities in the relation between heritage and city is the emergence of urban cultural tourism. One need not look to instances of high-pressure or difficult-to-manage cities (such as Venice or Barcelona) to confirm the capacity of these factors to generate complex urban processes, whether from a physical, social, cultural, or economic point of view. The emergence of what many have critically called the "heritage industry" has never been far removed from these processes.

This complex relationship between heritage and tourism, both theoretical and applied, was joined in the final decades of the 20th century by the appearance and generalization of various computer-based visualization technologies which have brought new possibilities in terms of the

registration and inscription of heritage, along with its protection, safeguard, and management, as well as the processes of transmission, mobilization, and promotion [1]. These aspects are here considered in relation to urban heritage, the historical city, and urban tourism in order to reflect on two computer-based visualization technologies of increasing significance to the management, presentation, and communication of heritage: augmented reality (AR) and virtual reality (VR).

As pointed out by Han and Jung [2] in relation to AR, but equally applicable to VR, although efforts have been made to achieve a commonly accepted definition, such has not been possible largely because these are developing technologies that have not yet reached their full potential. Regarding VR, Guttentag [3] defines it (using various sources) as "the use of a generated 3D environment—called a 'virtual environment' (VE)—that one can navigate and possibly interact with, resulting in real-time simulation of one or more of the user's five senses" [3] (p. 638). Among its characteristics, Gutierrez, Vexo and Thalmann (2008) [4] point out as essential elements of VR an "immersion" related to the physical and material (ranging from "fully immersive" to "non-immersive"), as well as a sense of "presence"—a subjective concept associated with the psychology of users [4]. The result is that, rather than experiencing the environment where s/he truly is, the user experiences immersion in a virtual world with which s/he interacts, and from which s/he obtains sensory responses [5].

For their part, Kounavis et al. [6] (p. 2) define AR as "a visualization technique that superimposes computer-generated data, such as text, video, graphics, GPS data, and other multimedia formats on top of the real-world view, as captured from the camera of a computer, mobile phone, or other device. In other words, AR can augment one's view and transform it with the help of a computer or a mobile device, and thus enhance the user's perception of reality and of the surrounding environment". Therefore, the main difference with VR is that while this "creates a totally artificial environment, augmented reality uses the existing environment and overlays new information on top of it" [7]. This implies the possibility of interacting with and manipulating the physical and virtual worlds at the same time, and of registering and connecting objects from both environments [2], thus integrating in real time the digital information with the live image of the environment in which the user is moving [7].

The difference between VR and AR in terms of the possibility of relating directly to the heritage environment being enjoyed, and the need to do so in situ in the case of AR, is of vital importance. In this sense, AR's potential for development within the tourism industry has been clearly recognized, to the extent that the sensation of interacting with the place visited—in what has been described as "social presence" [8]—is more powerful than with VR. As a consequence, while VR remains an essential tool for the safeguarding and conservation of heritage, most analyses on the application of computer-based visualization technologies for the valorization of heritage and cultural tourism have focused mainly on AR, at least in the recent years.

Notably, attempts are being made at reaching consensus on how best to use these technologies to reflect, disseminate, and investigate cultural heritage. In this sense, the 'London Charter' establishes a series of principles to ensure that digital visualizations of heritage be "at least as intellectually and technically rigorous as long-established cultural heritage research and communication methods" [9]. In addition, such consensuses aim to cover uses ranging from those more traditionally focused on research and conservation (academic, educational, curatorial) to more commercial uses linked to the entertainment industry and tourism, all of which represent the dissemination of cultural heritage. In this sense, there are several works [2,10] that have researched the user requirements that will move consumers to use these technologies. According to Han and Jung [2] these would for urban cultural tourism include: simplicity, relevant and updated information, speed, safety and security, accessibility, social functions, personalization, power efficiency, context-awareness, and reliability.

In general, the consulted bibliography reveals an evident consensus regarding positive evaluation of the effects that AR and VR (together with other computer-based visualization technologies) present, both in the recovery and conservation of heritage and in its implementation for tourist value. From a strictly protectionist perspective, these tools favor "digital documentation, and can be helpful for conservation and restoration actions, simulating different solutions and determining the level of

intervention (...) testing different restoration hypothesis at a digital level" [1] (p. 396). However, these same authors point out that the possibilities go much further and highlight the important role they can have in achieving the Millennium Goals and sustainability in relation to cultural heritage, both in terms of protection and safeguard and for promotion via tourism [1].

In fact, the potential for tourism is evident to the vast majority of authors. In the urban context, they are perceived as a potential tool to overcome the physical limitations of tourist attractions, and (to the extent that they use the digital space to offer information and additional value) as a facilitator in the sustainability of the heritage spaces [2]. This at a time when cultural tourism is changing the focus of "informative enrichment of cultural products to the experience of cultural heritage" [11] (p. 70). In this context of social and cultural change in tourism consumption, the aspect highlighted by most authors is the possibility of creating richer and more immersive content that increases consumer satisfaction [12] by enhancing the tourist experience [1,11,13–16].

In any case, and apart from the satisfaction of consumers, the value perceived by the different stakeholders in the use of AR for heritage tourism is much broader, as pointed out by tom Dieck and Jung (2017) [17]. According to this research, these values focus on aspects that are economic (attracting new target markets, justification to charge admission, incentives to return, etc.), experiential (interesting and interactive experiences, enriching memories, etc.), social (gamification, sharing experiences, social fulfilment, etc.), epistemic (new concepts for engagement, increasing attention when using an alternative approach, curiosity, etc.), historical and cultural (adding more content, telling personal stories of past events, triggering interest in history, etc.), and educational (personalized learning experience, learning at one's own pace, saving content for later, etc.). In general terms, there is a broad consensus regarding the importance of these values, except perhaps in some of the economic aspects, to the extent that some authors point out that the financial implications and income model for the implementation of AR projects in tourism are not yet sufficiently clear [14].

The cited bibliography represents only a small sample of the large number of works developed in recent years that reflect on the impact of AR and VR on the enhancement of heritage, cultural tourism, and the valuation of users. These are essentially empirical approaches, which (except in very exceptional cases) maintain an optimistic tone, clearly focusing on the benefits provided. However, there is another line of reflection, much less voluminous but of longer tradition, which from a clearly theoretical point of view has placed the focus on changes—in some cases negative—that computer-based visualization technologies and "virtualization" [18] can produce on our conception of the historical urban space as the essential locus of socialization, and of heritage as a discursive construction that is subject to conflict.

In effect, accepting that any application of these new technologies to heritage would be developed via "scholarly rigour", as the aforementioned London Charter [9] prescribes, what cannot be elided are the enormous possibilities that they offer in terms of presentation of the "(c)hange over time, magnification, modification, manipulation of virtual objects" [9]. Nor will they cover the way in which we relate to heritage and the urban space in an intellectual manner. Therefore, although the 'London Charter' is indeed a relevant bibliographic reference for this research, its main objective in "ensuring the methodological rigour of computer-based visualization as a means of researching and communicating cultural heritage" [9] differs from our own, more focused on analysis of the substantial changes that the generalization of AR and VR can entail in the social use of historical urban centers, and in what we decide to conserve, how we do so, and for what reason.

Thus, as indicated in greater detail in the methodological section, several empirical studies have been selected that include case studies assessing a specific AR or VR project around urban heritage. Next, a series of analytical categories have been developed in order to perceive whether the users of these technologies are aware of the benefits pointed out above as well as other processes, perhaps more subtle but no less important, such as possible changes in the sociospatial relationships that seem implicit and that underpin our understanding of what heritage really is. This we say without denying evidence (supported by practically all of the literature) that "tourists generally had a positive response on the use of AR for the enhancement of the urban heritage tourism experience" [2] (p. 5).

It is important to mention that, for this work, case studies have been chosen that evaluate projects concentrated on heritage in urban environments understood as 'spaces for the daily lives of inhabitants', leaving aside the more numerous analyses that focus on individual buildings, whether these be museums or closed archaeological sites. The reasoning here is that one point of maximum conflict that may involve the generalization of these technologies is the understanding of heritage as a discursive construction, subject to controversy, as well as the "virtualization" of public space as an area of socialization. These aspects can be more clearly perceived in living urban spaces (that is, which remain part of our present), regardless of their importance to the past, thus leading us to avoid spaces that are conceptually considered separate from said present, as with archaeological areas or museums.

Therefore, the hypothesis that supports our approach is that within still-living spaces of an historical city, the use of these new technologies will represent not merely a new form of documentation, intervention, representation, and transmission of the urban heritage, but may lead in a certain way to changes in how we personally relate to the past vis-à-vis the present. Thus, the material remnants that we choose to categorize as heritage may (through virtual and augmented representation) forge new relationships between categories of the past, present, and future within urban space.

To test this hypothesis, in order to determine whether the indicated changes are being produced and to document them, we should first consider in greater depth the concepts of heritage and urban heritage, as well as the spatial and temporal relationships that sustain these concepts, and that may ostensibly be transformed through the generalization of AR and VR.

## 2. Background

Following R. Williams [19], the classification of something as "historic" or "historical" requires the qualification of a sense of process or destiny and value. Indeed, while in Spanish (the native language of the authors of this study) there is only one adjective to describe this (histórico), the meaning in English can be split into two close (but not identical) terms—"historical," related to the study or representation of the past; and "historic," important (or likely to be so) in terms of history. Thus, a given element, apart from having a past, may be considered worthy of preservation and awarded meaning in the present. Further qualifying something as "heritage" in general, or as "urban heritage" in particular, implies historical significance as well as (more importantly for us) worthiness to history: that is, something to preserve to help build a certain narrative about a past to which it belongs, whether clearly or less certainly.

In general, this new assessment of the past and its links to the present can be connected to a certain socio-cultural conception of time, and to the existence of a "sense of historicity and singularity of the past [that] is an invention of the 19th century" [20] (p. 40). Thus, we are dealing with a phenomenon linked to Western European modernity, which later became universal. It should also be noted that these assumptions have remained fairly stable over nearly two centuries. Notably, the concept of heritage has become increasingly complex and is now applied to many more sorts of elements, both material and immaterial. Still, conceptual assumptions remain relatively similar to those implied in the initial conceptualization.

We consider what this conceptualization of heritage implies from a theoretical point of view then move on to question the possible effects of the generalization of AR and VR. Practical consequences are pointed out, but greater attention is paid to possible theoretical and conceptual changes that may involve a transformation in the way we define "heritage".

### 2.1. The Historic City as Cultural Heritage Selected from the Past to Suit the Present

Systematizations of the ideas of urban heritage and the historical city are linked to the general evolution of ideas relating to heritage and conservation. Following Choay, and Choay and O'Connell, [21,22], the terms "antiquity," "monument," "historical monument," and "heritage or historic city" highlight how Western societies view their relationship with time [22]. According to these authors, the first of these terms to appear (during the Renaissance) was antiquity and, since then,

these concepts have "played the role of a reflexive mirror (that) creates an effect of distance, opening an interval into which the referential time of history would insert itself" [22] (p. 138).

Although the concept of heritage as understood today has expanded over time to include more elements, it was systematized between the 18th and 19th centuries. However, the idea of a city being historical (and therefore heritage) developed only "towards the end of the 19th century and during the first half of the 20th" [23] (p. 38). This delay in the identification and assessment of a city and a set of urban processes as a complex heritage derives from the traditional consideration of a monument or work of art as a discrete and meaningful element (that is, something that begins and ends at a given moment, and contains a meaning granted by its creator from the beginning—a meaning considered to be unchanging over time) [22]. It was not until the first half of the 20th century that the city itself—a collective and perpetually unfinished construction—came to be considered at the same level as a monument and, in this sense, was protected as a heritage element.

The concept of heritage, as stated, entails two important facets. The first is the temporary separation between the present that we live in and a "past other," from which a series of elements of very different character have been inherited. Consequently, the idea of "otherness" is inserted into heritage from the beginning, to the extent that we must operate from a caesura—a point at which something may be considered to come from a past different from our present. As has been indicated, part of the city came to be understood as heritage in the 19th century, but this was not consolidated until the 20th. It was then that part of the city came to be perceived as a "past other" city, in opposition to the contemporary city, in the process of being built. The points of rupture between past and present were the profound transformations undertaken in cities during the 19th and early 20th centuries to adapt them to new industrial needs, and supposed the destruction of many older urban structures, along with profound social and economic changes [23–27].

As Hartog has pointed out, the historic monument category "presupposes that a certain gap has been opened: A moment comes when a monument can be regarded as something other than what it had been for a long time" [24] (p. 156). However, the temporal assumption underlying the idea of heritage may be changing due to computer-based visualization techniques, since (as Siberman points out) "the past has become an ever-present virtual reality that is simultaneously more real and more virtual than ever before" [12] (p. 9).

The second of the processes operating from our conceptualization of heritage is that of selection. The individual or spatial set of elements categorized as heritage are not determined by the past but are selected from the set of known material and immaterial elements rooted in that past. Consequently, the past is itself less relevant than the manner in which these elements and traditions have been selected as resources for the present. They are chosen and interpreted according to demands of the present to such an extent that they have as much to do with what, at each moment, one decides to remember as with what one decides to forget [28].

It is one thing for certain events to have occurred in the past and left traces, but quite another to be able to know that past only through a process—first, of selection, and then of discursive registration in a specific narrative about that past [29]. Indeed, the selection of certain elements of the past and their consideration as heritage is more than a simple task of conserving remains—it is the construction of a narrative about the past in relation to the present and the future. This also entails, among other aspects, a rejection of the assumption that an entire society must identify with a specific and coherent heritage that articulates the collective memory of its inhabitants. In this sense, the concept of "dissonant heritage" developed by Tunbridge and Ashworth [30] is of great interest. They point out that "all heritage is someone's heritage and, therefore, logically not someone else's: The original meaning of an inheritance (from which 'heritage' derives) implies the existence of disinheritance, and, by extension, any creation of heritage from the past disinherits someone, completely or partially, actively or potentially. This disinheritance may be unintentional, temporary, of trivial importance, limited in its effects, and concealed; or it may be long term, widespread, intentional, important, and obvious" [30] (p. 30).

However, this first dissonance, deriving from the process of selection (or omission), is joined by the second derivative of its "tourist commodification." What may, for one social group (in a more or less consensual way), be considered places of testimonial or cultural importance—or even sacred places—may become for others mere product-places that are continually sold, interpreted, and consumed (Graham, 2002) [28].This aspect has taken on immense importance for the city given the explosion of urban cultural tourism and what has come to be called the heritage industry—an "obsession with historical nostalgia that swept through ... Western Europe in the final years of the past century" [31] (p. 1098).

The heritage industry has unquestionably fueled one of the greatest economic forces of recent decades through tourism in general and urban cultural tourism in particular. Historic and heritage cities have come to attract special tourist attention, thus making it necessary for us to be cognizant that this tourist outlook has been highly determined [32]. If heritage implies a selection of elements from the past based on values in the present, tourism further implies a second selection from among these elements that, according to different variables, have the capacity to attract touristic attention (thus directing greater benefits, to whoever the beneficiaries may be). Consequently, tourism and its processes (marketing, commercialization, stories, guides, brochures, photos, visitors, queues, etc.) generate a new narrative and a new "regime of historicity" by way of the elements selected to attract tourism.

As some authors have pointed out, when complemented by AR and VR technologies in which fiction prevails over reality, another selection process is added to (or replaces) the experience, involving different criteria. This new choice includes only "what is strictly necessary to complement the virtual world" [33] (p. 252) and a particular narrative mediated by technological aspects. Along with the numerous avenues opened by such new technologies, there is the risk that (from the point of view of our understanding of the urban cultural heritage and the historical city) AR may become a "diminished reality" [33] (p. 252)

This should lead, inevitably, to a questioning of the objectives behind the construction of these new "regimes of historicity" via AR and VR, and which in the case of tourism and its derivatives are, as Silberman points out, essentially "lucrative" [34] (p. 9). If the objective of a large share of heritage presentations (especially those used in new computer-based visualization technologies, characterized by their complexity and high cost) is to attract consumers, it is obvious that these "can rarely afford to offer the kinds of serious and troubling historical reflections that are likely to drive holiday visitors away" [34] (p. 85). The result in such a case would be the engendering of a new and strictly contemporary perspective on the past, "bringing it dangerously close to being a state-sponsored commercial enterprise" [34] (p. 84).

Consequently, it seems important to include in this discussion the concept of authenticity in tourist-heritage representations, especially if we refer to contexts such as urban spaces, which under new tourism trends have ceased to be presented as places with monuments and traditions to be sold and consumed as authentic "heritage experiences". For this reason, "visitors carry some expectations that those experiences are, at least to some degree, realistic, accurate, and authentic" [35] (p. 213).

It is impossible to discuss here at length the idea of authenticity—possibly among the most complex and controversial in relation to the present debate. Note, however, that the interpretation is being consolidated that authenticity is not only a subjective quality, but also a relative one, extending to different types (such as "original authenticity", "authorized authenticity", "perfect reproduction" or "authentic reproduction") [35,36] or to different degrees ranging from the totally authentic to the totally inauthentic [3].

The incorporation into this debate of technologies such as AR and VR does not appear to change the theoretical terms of the discussion, beyond offering certain commentators the illusion that it may finally be possible to overcome the resistance of the past to be faithfully represented. However, as Silberman points out, the past "is one of the most virtual of the realities we have to contend with" [34] (p. 9), which should lead us to "resist overstating the potential of digital heritage for creating a definitive, objective reconstruction of the past" [34] (p. 83).

We should, therefore, avoid the temptation to present augmented or virtual recreations of the patrimonial spaces (whatever their objectives: tourist-economic, educational, declarative, reconstructive, curatorial, etc.) as an authentic and definitive presentation. This does not mean that technologies such as AR and VR cannot suppose a huge advance for reflection, as long as we bear in mind that these are not neutral representations of a supposed past reality but instead respond to the particular interests of specific agents, thus regarding heritage as a complex discursive narrative, subject to different approaches, perspectives, and power relations. As Silberman concluded, "we should not think only of creating more cleverly packaged products, but instead create information avenues of two-way communication, in which ( . . . ) alternative visions of the past can make themselves heard" [34] (p. 12).

### 2.2. Places of Memory and Virtual Places of Memory: A New Acceleration of Time?

Boyn [20] has pointed out that the propagation of the concept of heritage in the 19th century cannot be understood without contemplating the feeling of nostalgia derived from the accelerating pace of industrialization and the longing for a vanishing time in which life moved slower, both in socio-cultural reality and physical materiality. As French historian Hartog points out, during the period in which heritage became a dominant, encompassing category of all cultural life, "it was treated as though it were self-evident" [24] (p. 150), when, indeed, "heritage makes visible and expresses a certain order of time, in which the dimension of the past is the most important" [24] (p. 152).

Thus, there arises an obsession with a past that is directly at odds with our ability to keep it alive, leading French historian P. Nora to develop his concept of "places of memory". This idea was developed in a moment of temporary crisis with the goal of "stopping time, blocking the work of forgetting, fixing a state of affairs, immortalizing death, materializing the immaterial (...) to enclose a maximum of meaning within a minimum of signs" [37] (p. 38). It is therefore an "acceleration of time" that produces the crisis at the origin of the place of memory and, as Hartog [24] contends, of our current obsession with heritage.

Considering all this, the question emerges of whether, in present time, we are not experiencing another moment of temporary acceleration and thus assisting the emergence of a new kind of place of memory that, in this case, can acquire a virtual character irrespective of physical materiality, with an enormous capacity to build new regimes of historicity. In this sense, we are interested in questioning not the technical aspects and possibilities offered by advances such as AR and VR, but rather the reasons for their expansion based on our understanding of past and present relationships, as well as their impact in terms of understanding urban heritage and the historic city. In any case, we should not disregard the long-term impact they may have on the physical and social materiality of the city, since (to a greater or lesser extent) they will eventually influence what we preserve, how we preserve it, and what future uses may be made of the heritage in question.

### 3. Methodology

In response to the theoretical issues raised in the previous sections, the fundamental objective of this study is to determine how the new AR and VR technologies influence our interpretations of urban cultural heritage and tourism. To this end, prior academic discussion on this topic has been reviewed through the analysis of those articles with the greatest scientific impact over the last 10 years (2007–2017).

An exploratory approach to the subject allows us to affirm that these technologies are tools that favor the transmission of information and knowledge and improve a visitor's experience. However, problematic issues can arise, such as whether the creation of virtual worlds might affect the spatial and temporal perception of the city and its cultural heritage, or whether they might drive a simplification of the touristic and cultural experience of the visitor by emphasizing the most playful aspects of the technologies.

To answer such questions and fulfill the proposed objectives, this study is presented in three phases in a methodological sense (Figure 1). These phases have allowed us to detect the most recurrent themes cited by experts and refine the information obtained through a selection of topics based on concurrences to sharpen the analysis and draw concrete conclusions.

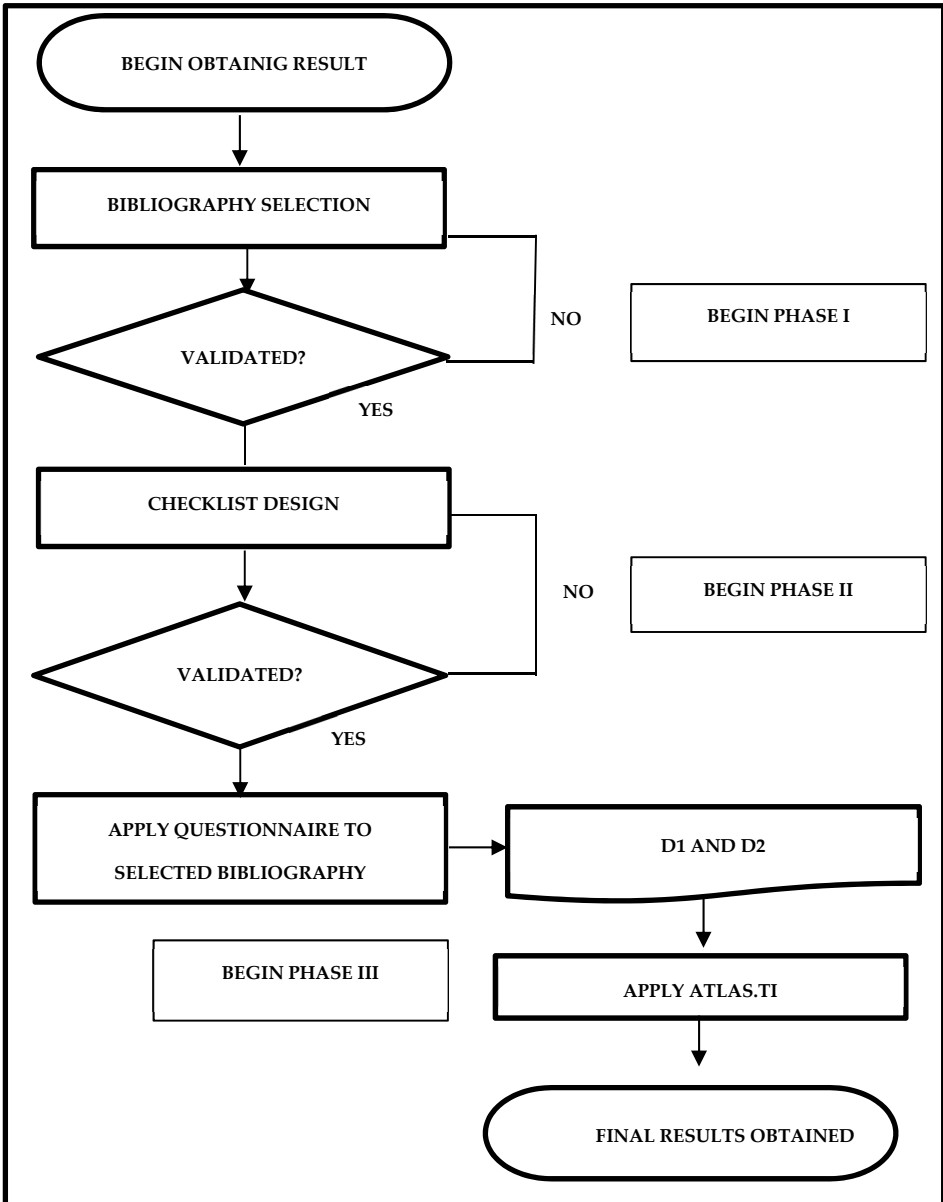

**Figure 1.** Flow chart. Obtaining data. Source: own elaboration.

In the first phase (Figure 1), a bibliographic selection was made of articles, conferences and proceedings that exclusively present case studies related to the use of AR and VR for the interpretation of the cultural heritage of historic city centers. This being a bibliographic investigation, we wanted to ensure the scientific quality of all sources used as well as the relevance of the content. Therefore, special attention was given to two elements: the academic and scientific impact of the selected publications, and the year of publication. In this sense, our thematic selection criterion was the discovery of case studies focusing on the interpretation of the cultural heritage of historic centers via AR and VR. Investigations that did not include empirical experiments developed in historical city centers were systematically excluded. This reference search was carried out through two international databases—Web of Science, managed by Clarivate Analytics; and SCOPUS, endorsed by Elsevier—and

yielded a total of 30 publications (22 articles and 8 conferences and proceedings) (Table 1 and Figure 2). The keywords used to perform searches were: "augmented reality", "virtual reality" and "cultural heritage". All sorts of journals were considered regardless of their nominal theme. As a testament to its scientific quality, we note that 27 articles were indexed in the Scimago Journal (SJR) and eight in the In Cities Journal Citation Reports (JCR), and that 44.4% of the articles ranked in the first quartile in established rankings (Table 2). At first, analysis of our research was carried out only through articles, with lectures and conference proceedings subsequently incorporated. The results obtained with this inclusion of more publications proved very similar, and thus it was decided to limit the sample size to 30 publications in total, according to the criteria used in such methodologies (Appendix A).

It is necessary to emphasize that the present research is not quantitative and does not use methodologies or tools of that nature. Regarding the size of the sample in qualitative research, Salamanca and Martín-Crespo [38], based on the criteria of Hammersley, Atkinson or Gumperz, state the following: "regarding the size of the sample, there are no firmly established criteria or rules determined on the basis of informational needs; therefore, one of the principles that guides the sampling is the saturation of data to the point where new information is no longer obtained, as it begins to be redundant".

**Table 1.** Scientific publications related to the subject in the 2007–2017 period.

| Year | Number of Publications | % |
|------|:----------------------:|:----:|
| 2007 | 1 | 3.3 |
| 2008 | 1 | 3.3 |
| 2009 | 0 | 0.0 |
| 2010 | 1 | 3.3 |
| 2011 | 1 | 3.3 |
| 2012 | 2 | 6.7 |
| 2013 | 2 | 6.7 |
| 2014 | 6 | 20.0 |
| 2015 | 4 | 13.3 |
| 2016 | 5 | 16.7 |
| 2017 | 7 | 23.3 |
| Total | 30 | 100 |

Source: Own elaboration.

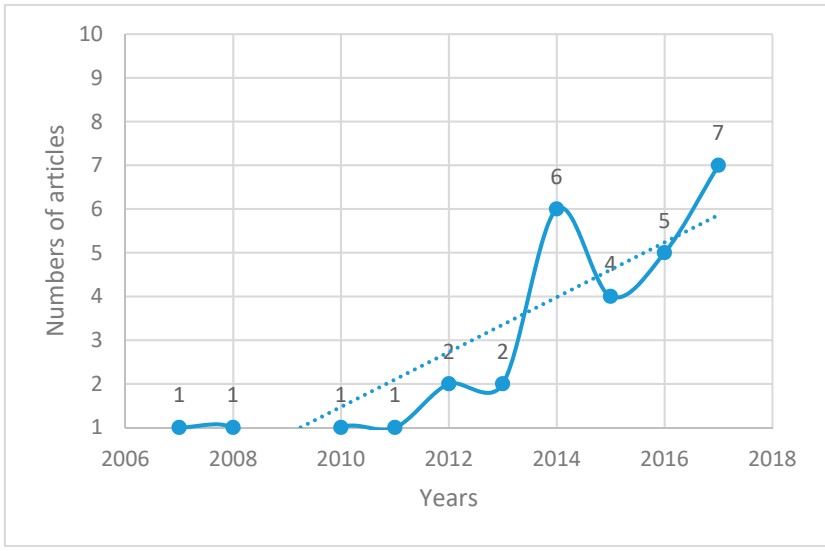

**Figure 2.** Scientific publications related to the subject in the 2007–2017 period. Source: Own elaboration.

**Table 2.** Bibliographic metrics of scientific publications selected.

| Quartile JCR | Frequency | % | Quartile SJR | Frequency | % | Frequency Quartile JCR/SJR | % |
|---|---|---|---|---|---|---|---|
| Q1 | 3 | 33.3 | Q1 | 8 | 47.1 | 12 | 44.4 |
| Q2 | 4 | 44.4 | Q2 | 2 | 11.8 | 6 | 22.2 |
| Q3 | 1 | 11.1 | Q3 | 6 | 35.3 | 7 | 25.9 |
| Q4 | 1 | 11.1 | Q4 | 1 | 5.9 | 2 | 7.4 |
| Total | 9 | 33.3 | Total | 17 | 100 | 27 | 100 |

Scimago Journal (SJR) and Cities Journal Citation Reports (JCR) indices. Source: own elaboration.

Both AR and VR have undergone constant and rapid development that quickly renders previously developed tools obsolete. For this reason, it was considered appropriate to prioritize in the selection criteria research articles published in the 2007–2017 period. The temporary sustainability of these technologies is one of the main concerns of the London Charter [9]; in fact, principle number 5 focuses completely on this concept: "strategies should be planned and implemented to ensure the long-term sustainability of cultural heritage -related computer-based visualization outcomes and documentation, in order to avoid the loss of this growing part of human intellectual, social, economic and cultural heritage" [9]. A detailed analysis of this historical series shows that the highest percentage of articles compiled for this study (73.3%) was concentrated in 2014–2017 (especially 2017, with 23.3% of the total) (Figure 2). It can be said, therefore, that the data were up to date.

Once this initial part of the investigation was concluded, the second phase (Figure 1), containing a marked experimental component, was launched. A checklist (questionnaire) was composed using the "Google Forms" tool with the aim of systematizing the main contents of the selected articles and conducting an analysis of their content. The checklist can be viewed at Supplementary Materials. A total of four general topics were established (Topic 1: Impacts of AR and VR tools on the cultural heritage of urban historic centers. Topic 2: Functionality of AR and VR tools applied to the cultural heritage of urban historical centers from the point of view of the user. Topic 3: Perception of the city through AR and VR tools. Topic 4: AR and VR tools and city stakeholders), further divided into 21 sub-themes (Topic 1: Conservation and protection of heritage; Dissemination and promotion of cultural heritage; Creation of touristic cultural resources and products; Development of nomadic museography; Ephemeral tools as a consequence of constantly developing technologies. Topic 2: Quick or immediate access to knowledge and information; Reinforcement of M-Learning theory; Social inclusion of disadvantaged groups/social exclusion through digital training; Virtual recreation versus virtual restitution; Sensitization toward cultural heritage; Empowerment of the heritage experience. Item 3: Simplification of enjoyment of the city; Enhancement of the heritage experience; Creation of symbolic worlds other than real ones; Loss of social and political functionality of the city; Unraveling with respect to the social reality of the urban environment; Banalization of social and cultural processes. Topic 4: Public-private collaboration; Development of smart cities; Entrepreneurship and innovation fora; Development of mobile technology (3G, 4G, and 5G); Inclusion in tourism, cultural planning; etc.)

These themes and sub-themes, widely developed in the state of the art of this article, derive on the one hand from scientific reflection by the authors in relation to how AR and VR can modify the traditional concept of cultural heritage, affecting space-time perceptions of the historical centers of cities. This reflection was based on the bibliographic review of authors who have likewise shown scientific interest in the impact of AR and VR on the appreciation of historic centers by users of mobile devices equipped with these technologies. Such is the case of Santamarina [33], reflecting on the creation of virtual worlds distinct from real ones; also, Urry [32], who points out how the use of AR and VR might simplify enjoyment of the city. On the other hand, a second round of topics and sub-themes were linked to aspects related to "M-learning", patrimonial education, digital technology, and cultural heritage. The process here was similar to the first case, with the authors undertaking deep

reflection based on the bibliographic review of experts in this field, such as Santacana and López [39] or Santacana and Coma [40], dealing with aspects of how "M-learning" can influence the acquisition of knowledge, or the impact of digital environments on contemporary theories of protection, conservation, and restoration of cultural heritage [3].

The checklist was applied to the compiled bibliographic corpus by the authors of this article to determine the topics addressed, as well as the approach expressed regarding the use of AR and VR in the valuation of urban cultural heritage, whether from a positive, negative, or ambivalent perspective. It is necessary to clarify that the checklist was not answered by any participant, but was conceived as a tool to help researchers detect and fragment information from the selected bibliography. This procedure for collecting and obtaining data was based on a bibliographic review of the 30 selected publications. Each article was analyzed by way of the checklist based on the 21 proposed sub-themes. These topics were assessed, also through the checklist, with a positive, negative, or neutral approach, depending on the perspective of each publication. The contents comprised two documents, one with positive aspects (D1) and another with negative and ambivalent aspects (D2), which were analyzed through the ATLAS.ti software.

However, prior to final application of the checklist to the selected bibliographic corpus, a pre-test was conducted based on a review of the first five articles, in order to validate the checklist. This pre-test involved the modification of topics that were deemed too specific and therefore inoperative. As a consequence, the following sub-themes were reoriented:

- Sub-theme 1.4. "Development of Nomadic Museography". Nomadic museography is defined as "the accumulation into a single portable (nomadic) device of various computer and communications functions that allow a new relationship between the user and museums" [41]. During the pre-test, this definition was found not to be very representative. It was decided therefore to propose a broader concept of nomadic museography in which the barrier of the museum as a physical space was overcome, extending through the historical cases of cities.
- Sub-theme 2.3. "Social inclusion of disadvantaged groups / social exclusion through digital training". As in the previous case, the application of digital training to disadvantaged groups offered no results, so this topic was analyzed taking as a reference the society as a whole, and not only disadvantaged groups.
- Sub-theme 4.3. "Entrepreneurship and focuses on innovation". No focus on entrepreneurship and innovation was detected during the pre-test, so this sub-theme was reoriented towards aspects related to the economic cost and dynamism that RA and RV technologies are capable of generating.

To facilitate the analysis and interpretation of information obtained during the above experimental design (D1 and D2), the ATLAS.ti software was employed in a third phase (Figure 1). This is a program centered on qualitative data analysis that allows the examination of large volumes of information while focusing on its content. The use of this software allowed us to determine connections, hierarchies, and existing networks between the different topics and sub-themes and establish results as well as systematized conclusions.

## 4. Results and Discussion

The thematic specification provided by ATLAS.ti allowed us to carry out a second, deeper and more detailed, bibliographic analysis of the selected texts. The most relevant topics were taken as a guide for study in which the positive, negative, or ambivalent relationships between urban cultural heritage and the new AR- and VR-based technological tools are emphasized.

### 4.1. Visualization of a Thematic Mosaic Based on the ATLAS.ti Code Concurrency Technique

Once the most commonly recurring themes in the selected bibliography were established by way of the checklist, the extracted information was qualified by determining concurrency relationships among the issues using the ATLAS.ti software. Subsequently, these topics were classified according

to 37 codes, 28 of which represented positive relationships involving AR and VR ("Gamification," "Territory management," "Identity," "Interaction with cultural heritage," "M-Learning," "Improvement of information," "Improvement of awareness," "Improvement of accessibility," "Improvement of conservation," "Improvement of visitor experience," "Improvement of protection," "Improvement of preservation of historical values," "Improvement of knowledge," "Improvement of management and planning," "Improvement of public–private collaboration," "Improvement of tourist destinations," "Virtual worlds and symbolic worlds," "Virtual museum," "Nomadic museography," "Intangible heritage," "Prevention of damages," "Digital projection," "Reactivation of the heritage," " Recovery of the vanished heritage," "Sensitization," "Smart city," "Economic Cost" and "Theory of digital restoration") and nine of which manifested negative and/or ambivalent links ("Gamification," "Improved knowledge," "Improved visitor experience," "Improved information," "Improvement of tourist destinations," "Improvement of management and planning," "Virtual worlds and symbolic worlds," and "Sensitization"). In the positive relationships, 75% of codes (a total of 21) showed concurrences or intersections with other themes (Figure 3). Negative or ambivalent codes, although clearly fewer, showed a higher percentage of concurrences, at 88.8%: that is, in eight of the nine coded topics (Figure 4). The degree of concurrence in both cases was very high, implying an intense correspondence between all subjects that had appeared. For more clarity on the contents of each of the codes used in this analysis, descriptions are provided in Appendix B.

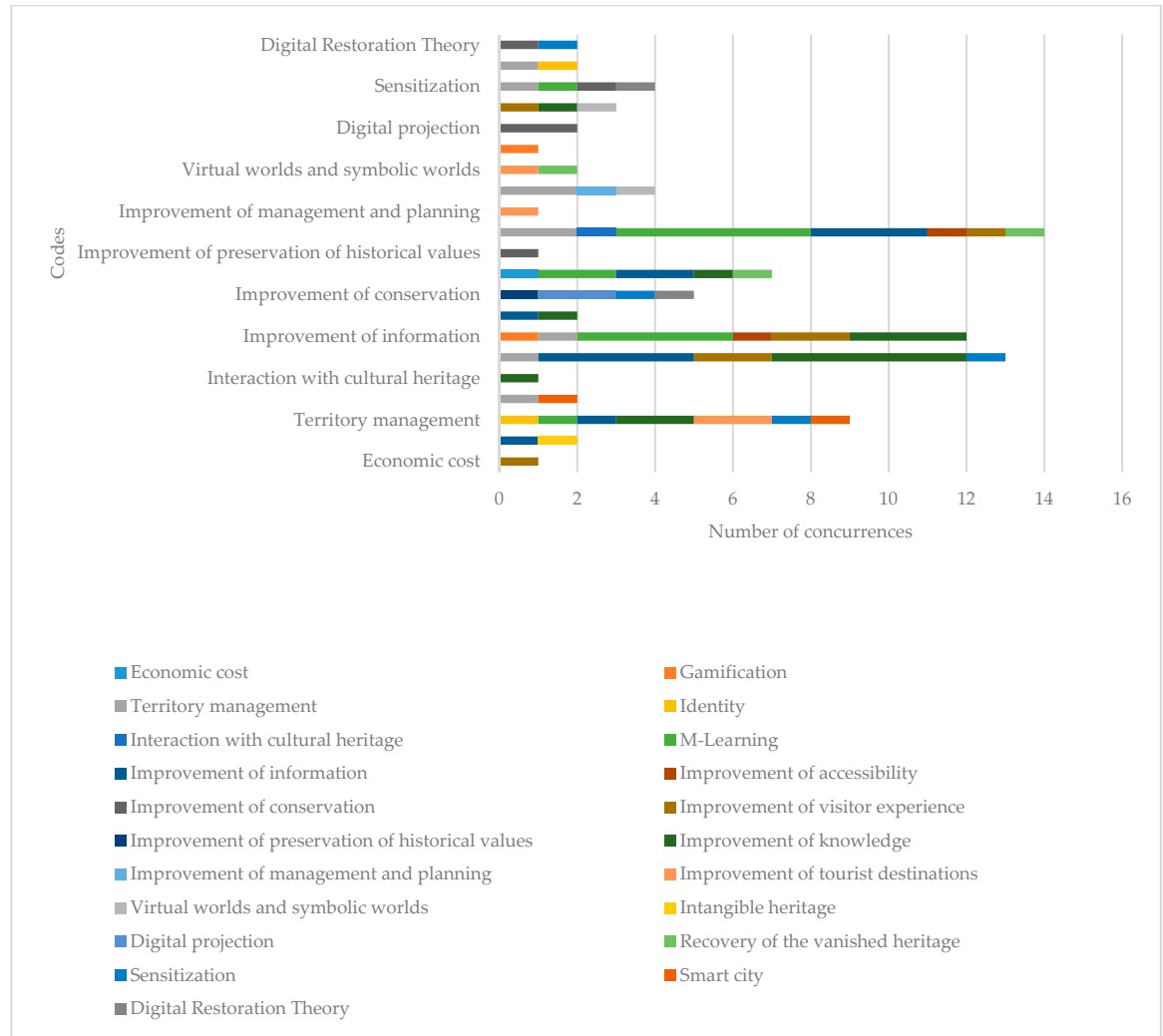

**Figure 3.** Mosaic of positive concurrency determined with ATLAS.ti. Source: own elaboration.

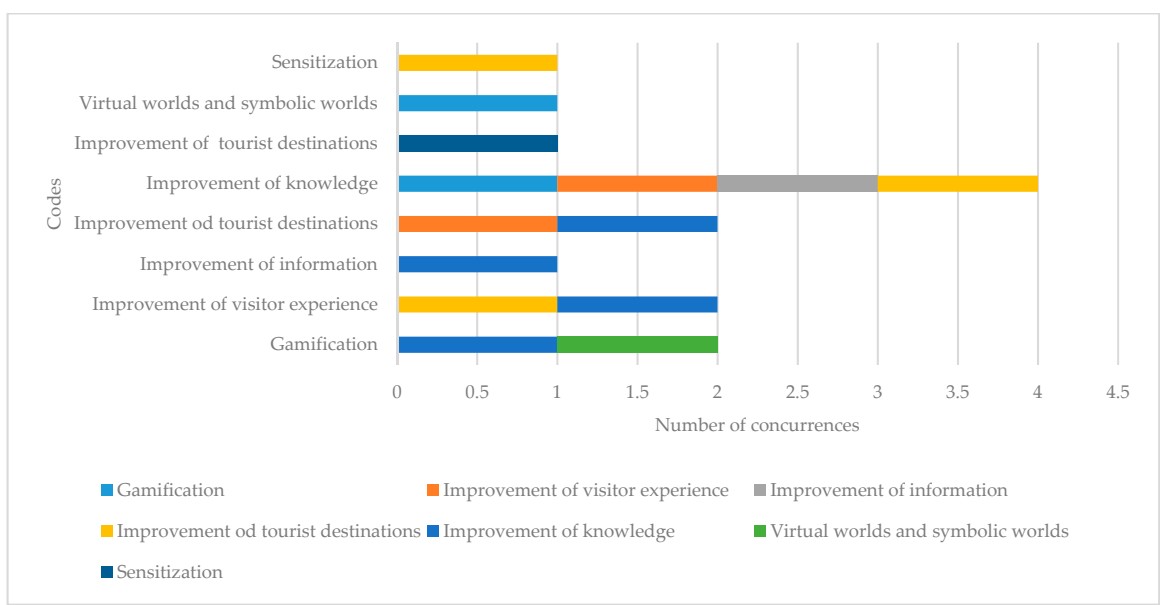

**Figure 4.** Mosaic of negative and ambivalent concurrencies determined with ATLAS.ti. Source: own elaboration.

Within the group that manifested a positive relationship with AR and VR, the codes "Territory management" and "Improvement of knowledge" represented the highest number of concurrences, with a total of seven, followed by "Improvement of information," with six intersections, and "M-learning," with five (Figure 3). In light of this result, together with the remaining codes and concurrences in this set, a bibliographical analysis of the texts was developed that focused on the opinions and assessments of scientists concerning the effects of AR and VR on cultural heritage in terms of access to information and the new forms of knowledge generated with these technologies, especially in the case of M-Learning and the novel opportunities that AR and VR offer to touristic management of cultural heritage.

In instances of negative or ambivalent relationships, "Improvement of knowledge" was the code with the greatest number of concurrences, with a total of four, followed by "Improvement of tourist destinations," "Improved visitor experience," and "Gamification" with two each (Figure 4). Considering these results, the bibliographic analysis of this negative grouping focused on two aspects: the geographical and temporal disconnections that the user may suffer when AR and VR are applied to urban cultural heritage; and the trivialization of the tourism process in terms of improving knowledge, information, and the visitor's experience.

*4.2. M-Learning and Tourist Management of the Territory. Positive Consequences Derived from the Use of AR and VR*

Our concurrency analysis found that "Territory management" was the code that intersected with the greatest number of topics from a positive point of view, as shown in the following diagram (Figure 5). The main intersections were produced with topics related to awareness, interpretation, and commitment to heritage by users; also with aspects related to the improvement of knowledge and patrimonial education; and with effects on tourism, both in terms of disseminating information to visitors and in relation to management of the activity and the creation of new products.

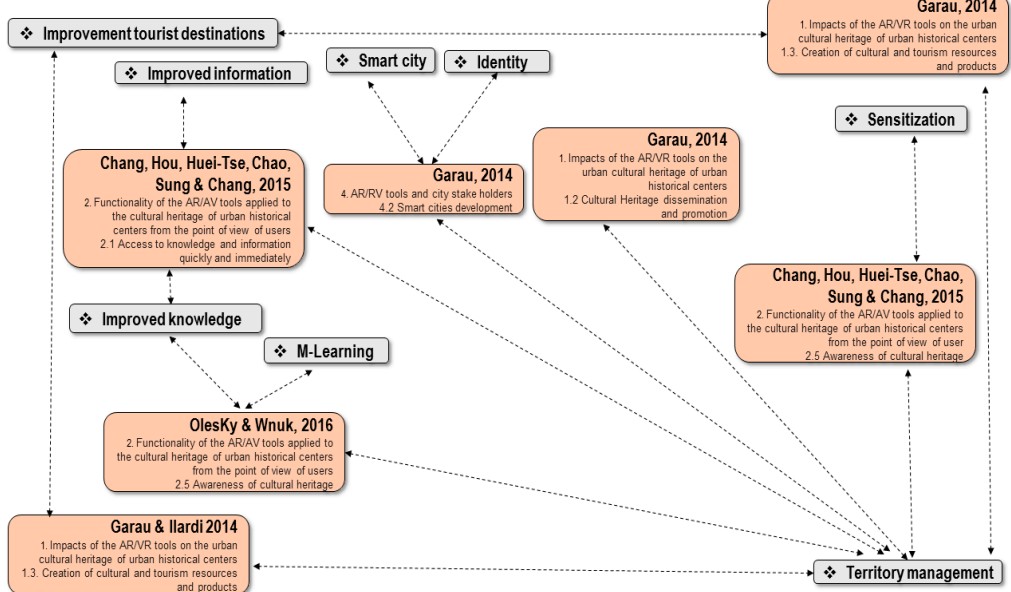

**Figure 5.** Networks of concurrencies of codes and citations in texts with respect to "Territory management". Source: own elaboration.

Most of the case studies analyzed in this study have highlighted the positive effects of the use of AR and VR in the assessment and interpretation of urban cultural heritage. Many of these effects are related to the cultural or touristic experience of users of AR and VR tools, especially in relation to new forms of access to information and knowledge (Figure 5), which are much more complex, interactive, and immediate than traditional channels [42–46]. Other authors see these technologies not only as a means for creating new products [42–46] and tourist destinations [47], but also for the promotion and dissemination of tourism products at different territorial scales or for the territorial management of the destination [48]. From the point of view of final users (tourists), these technologies would serve those who are preparing for a visit—favoring, for example, decision-making around which destination to choose [3]—as well as those who have already visited the destination [44]; but also those who cannot visit, being unable to "overcome the space, time and location restrictions that might occur at the physical site" [13] (p. 104). In conclusion, the technologies can generally allow for "enriching the tourist experience, adding information to the real environment merged and over the device display" [49] (p. 122).

In a still more general sense, for tourists as well as for locals who uphold the urban cultural heritage, these types of technologies can strengthen the identification of a society with its own territory, increasing the urban user's awareness of the surrounding cultural heritage [13,50,51]. As a result, VR and AR can allow the city (or the wider territory) to assume the role of an "open museum" [35] (p. 66), and new technologies within the broad context of "smart" city or region "can be used increasingly for touristic and cultural purposes" [52].

Thus, AR and VR can improve information and knowledge concerning a destination, and furthermore, as numerous authors have pointed out, such tools can also favor the development of "M-Learning," [53–55] and strengthen the cultural experience of the user by generating new ways of relating to cultural heritage [15,40,42,45,49,56,57]. For example, a study by Olesky and Wnuk [43] concludes that AR's ability to evoke a city's past by way of overlapping images is an excellent means of transferring knowledge; and Hain, Löffler and Zajícek [58] point out how technologies can recreate in real time the atmosphere and spatial characteristics of a chosen time period, including animations or interactive sound effects that make a strong impression on users and allow them to "rediscover history in a new and exacting way" [58] (p. 2034).

The ability of AR and VR to offer deeper experience and knowledge to a visitor is generally agreed upon by experts, and many authors have emphasized how these technologies enhance the experience of heritage [13,15,41,42] "promoting the sense of belonging to the site" [13] (p. 104) and, consequently, favoring both conservation and enjoyment of the heritage site. Garau and Ilardi [59], for example, underline the immediacy of these technologies, and that they favor cumulative knowledge. Malegiannaki and Daradoumis [48] argue that AR and VR generate close experiences in which users can create their own narratives, which causes them to develop more emotional reactions and even change their attitudes toward the subject matter. This reflection was developed by Hincapié et al. [60], who claimed that the benefits of the digital world can change values, attitudes, and skills in relation to heritage. Moreover, according to Mortara et al. [56], interaction through AR and VR increases involvement with heritage, especially in terms of learning; but it also reinforces feelings of simple enjoyment given that tourists can better understand historical facts and better appreciate heritage, especially elements that no longer materially exist. Further, many AR and VR platforms allow users to generate collective knowledge that can change how we create and share knowledge, although such ostensible effects have not yet been evaluated in depth. In this sense, in response to certain criticisms that these technologies can promote user isolation, it is argued that they actually encourage shared experiences, because "multiple users can ( . . . ) log on simultaneously, socialize and create groups and communities through various communications channels" [44] (p. 481).

One of the more interesting aspects of the interpretation of urban cultural heritage pointed out by researchers is that of gamification. The positive effects of this technique have been accepted by experts, especially in the case of "serious games:" that is, games that treat cultural heritage with historical and scientific rigor. Mortara et al. [56] summarized these benefits very well in their study on AR and mobile devices: greater involvement by users, reinforcement of knowledge, dissemination of missing or partially preserved heritage, etc. Two particularly beneficial effects for the protection of heritage should be highlighted: "cultural awareness" and "heritage awareness," which promote users' sensitivity to heritage and culture, and increases involvement in both cities and museums [56].

The use of these technologies applied to the interpretation of cultural heritage can attract new groups not currently interested in such aspects or assist groups that (due to issues such as disability) have been unable to access the heritage element [15]. It can even change the way information is constructed and disseminated, as users of technology may experience a kind of participation, unlike in the traditional top-down model where only experts may create and distribute information [59]. In short, new forms of collaboration can be generated, not only between public and private agents, but also between experts and the general public [45], and this may further contribute to the development of local communities by providing them with new tools of participation and cooperation [52].

In addition to new avenues for accessing information and knowledge, the researchers being studied have sometimes focused on topics that, although perhaps less correspondent, may also be relevant to an in-depth understanding of the relationships between AR and VR, and cultural heritage, such as the ways in which these tools can influence the protection and conservation of cultural heritage [15,60,61]. Certain authors have highlighted the roles that AR and VR can play in digitally reconstructing cultural heritage that has not been preserved [60] or which is at risk of vanishing altogether [62], including aspects such as the memory of a particular ethnic or cultural group [57], or certain intangible elements of heritage [46], or those under threat by armed conflicts or natural disasters [44]. This is not related to recreating symbolic worlds but virtual scenarios of cities that might have existed if structures of historical relevance had been preserved [43,51,63]. From another perspective, the virtual recreation of sites offers an alternative to destinations that; owing to their remoteness, physical fragility, perilous conditions, or mere nonexistence; prove inaccessible and/or challenging to navigate from a tourist's point of view in the real world [3]. These ideas coincide precisely with the principle number 6 ("Access") of the London Charter [9], which states that "the aims, methods and dissemination plans of computer-based visualization should reflect consideration of how such work can enhance access to cultural heritage that is otherwise inaccessible due to health and safety,

disability, economic, political, or environmental reasons, or because the object of the visualization is lost, endangered, dispersed, or has been destroyed, restored or reconstructed" [2].

For their part, such authors as Fang [47] have offered a more technical perspective on the conservation of cultural heritage by transferring orthodoxy from the theory of restoration to the field of digital restoration. According to Fang [47], digital reconstruction is achieved through the imitation of features that are already known and would allow, for example, virtual reconstructions that exceed the technical limits imposed on a given intervention [46,60]. In a much more general observation, Marques et al. [49] conclude that these technologies provide information that can be extremely valuable for all the agents involved, thus facilitating decision-making around heritage.

### 4.3. Spatiotemporal Disconnection and the Trivialization of the Tourist Experience: Negative Consequences of the Use of AR and VR

From a theoretical point of view, several authors have considered the possibility that AR and VR can subvert sociospatial relationships on which our concept of heritage is based, as noted early in this text. On the one hand, a contingency arises wherein the "technological progress erases the borders between reality and virtual reality. Perception of the world can be manipulated through the technology. Various illusions can be fabricated in the real world through the physical installations or in the mixed reality" [58] (p. 2032). However, this perspective, while indicated by the author theoretically, does not appear to be a wholly negative assessment, nor is it subsequently cited in the case study under analysis [58]. It should be noted that other authors question from the outset the danger of loss of value in terms of the socio-spatial reality of the heritage, noting that this "must not be regarded as a debasement of the real assets, but rather an added value and incentive" [44] (p. 483).

On the one hand, authors who empirically analyze the use of these tools (such as Chang et al.) [43] assume the possibility that they can undermine the geographical connections of heritage, not through confusion between the real and the virtual, but rather through disconnection between the material social space (locus of the heritage itself) and the virtual space. Indeed, this might occur if users focus exclusively on the relevant AR and its contents: "the limitations ... for AR mobile-guided activities lie in its inability to balance a visitor's attention-distribution between the information provided by the AR and the physical scene, causing them to focus excessively on the human–computer interaction and ignore the more important human-context in the real environment" [43].

This separation between material and virtual patrimonial spaces is sometimes raised in relation to the very concept of cultural tourism [3]. The essential question, in the case of computer-based visualization technologies not used within the represented place, but rather at a different location, would be whether this can still be considered a form of tourism or merely entertainment [3]. Although the empirical response is by no means overwhelming, some experimenters have noted that AR applications "might result in a missing out on real-life experience, as discovery is an important element of the tourist experience" [42] (p. 163).

This complex theoretical reflection is seen in case studies mainly in relation to tourism and the concept of authenticity [3,64]. In any case, it is clear that, whether in relation to tourism in general or heritage tourism in particular, not only is the concept of authenticity negotiable and socially constructed [65], the commodification process around a supposed "authentic heritage" cannot be seen as inherently positive or negative. It has, however, given rise to phenomena of enormous socio-economic complexity [66].

Another aspect derived from the translation of the sensitive and complex urban experience (physical, social, and political) to a virtual experience is the possibility of inherent simplification. In effect, AR and VR can conceivably collide with the construction of an "augmented fiction built on a reduced and simplified version of our material and symbolic world, which ceases to be the central object of experience and becomes just another intermediary for playful contingent acts" [33] (p. 257). Thus, AR could end up replacing an urban reality that is undoubtedly more complex from the sensory perspective—"more corporeal" in Virilio's words [18]—than a virtual experience that is almost entirely

visual) [64]. This would enhance what Urry identified as a problem of heritage: its emphasis on visualization, after which "various kinds of social experiences are in effect ignored or trivialized" [32] (p. 102).

Most empirical studies have focused on the case of tourism as a complex sensory experience and the simplification entailed in replacing travel with an essentially timeless spatial experience inside a "virtual environment." D. Guttentag [3] analyzed technological advances aimed at reproducing experiences beyond the visual (auditory, olfactory, tactile, and even taste). However, these advances suggest "there is still the dominance of the visual aspect of the immersive experience" [64] (p. 56), and that from an empirical point of view, "many aspects of the tourist experience may never be fully replicable" [3].

The simplification and possible "banalization" of social and cultural processes seen to accompany the heritage industry are not exclusive to the current technological revolution. These perils inevitably underlie almost all tourism insofar as this is an activity based on enjoyment and leisure (that does not necessarily exclude knowledge). Despite the undeniable advantages of new technologies, it must be noted that the concept of gamification when applied to VR may reinforce the playful aspects of "using" urban heritage, to the detriment of its educational, cultural, or scientific aspects [64]; and this danger would be accentuated were such developments left in the hands of the gaming and entertainment industries [58]. This very concern appears in various empirical studies, albeit with varying clarity. However, the final evidence seems to be contrary to the indicated concern. Chang et al. [43] seem to conclude that patrimonial guidance via AR does not lead to the banalization of the urban heritage experience, but may in fact improve the reception of knowledge compared with traditional guide systems, favoring a sense of place that fosters identification with the visited space and its values Chang et al. [43].

Regardless of theoretical aspects, from a practical point of view, the uprooting permitted by VR and AR can jeopardize positive economic effects that, like tourism and the heritage industry, are currently territorialized. In this sense, these technologies can undermine traditional discourse on the role of tourism and heritage in regional development. This problem was scarcely mentioned in the texts analyzed, and only in relation to tourism and its possible negative economic consequences for destinations in the sense of reducing the number of visitors [3]. In this regard, Soliman [46] notes that "virtual heritage threatens tourism itself, a source of income for many countries. A tourist may conceivably visit a virtual site or museum, without travelling to a real site" [46] (p. 90).

Finally, another aspect to consider is that access to these tools remains far from universal [42,52]. The socio-cultural construction of memory and heritage is not necessarily democratized by the use of VR and AR, but may remain confined to only people with greater social, economic, and symbolic power. In this sense, the issue of the cost for users is one of the most frequently cited negative aspects [13,44,46,51]. Even so, some authors argue that such costs can be mitigated by the economic benefits they generate [44], and that portability and ease of use potentially opens the enjoyment of heritage to new users [51,54].

In empirical analyses that address this concern, the problem is perceived to lie with the way in which information is transmitted through AR and VR (which may maintain a top-down direction, perhaps enhanced by technological complexity and associated costs) [47,59], as well as in terms of the new potential relationship between the public and private sectors [42,52]. In fact, a number of texts warn of the dangers of technological concentration and, consequently, that production of knowledge may be promoted from the top down in construction of the patrimonial discourse; these advocate for greater democratization with proposals such as the exchange of technological knowledge (as by international organizations) among countries with different levels of development [46], or that sufficient economic support be provided in order to favor entry and participation by different agents [51]. However, from a purely empirical analysis, it may be concluded that these theoretical dangers do not seem to be noticed by end-users.

Alongside aspects evaluated as positive, negative, or ambivalent consequences of using AR and VR in the interpretation of urban cultural heritage, further reflections can be incorporated. Although they do not register many thematic concurrences, they may propose future lines of research or merit greater study given the possibilities they open.

As noted earlier, the concept of heritage implies the establishment of complex space–time relationships. One may therefore wonder therefore whether, according to the experiments being analyzed, AR and VR have the capacity to modify these relations and subvert the traditional understanding of heritage from a geo-historical perspective. This might make a reality of Paul Virilio's late 20th century prediction that "the problem of virtual space is the loss of the real city" [18] (p. 46). This topic does not appear in most selected articles, but when it does, the authors tend to treat it with caution, without advancing strong opinions one way or the other. It should further be noted that while some authors point to the possibility of modifying traditional spatial relationships implicit in the concept of heritage, this does not appear to have been significant in the experiments carried out involving end-users. That is, while this may be a theoretical issue of interest to the academic world, its practical effects have not been examined among users of urban space and heritage.

C. Garau and E. Ilardi [59] took the above consideration a few steps further when they signaled the appearance of what they call "neo-places"—places "that are produced by the new digital media environment" [59]. They do not consider this to be an intrinsically negative process but do assume a compromise between what they describe as "old places" (the traditional patrimonial space) and spaces exclusively designed for consumption and transit, which anthropologist M. Augé dubbed "non-places." These "neo-places" are "where personalization (consumerism) and enduring social connections (of citizenship) find a balance, a junction where individual memories can connect to broader historical narratives so that the 'old places' can be rediscovered as heritage to reuse, protect, complete, and contextualize" [59].

In any case, as noted earlier, none of the studies that delved into this topic found empirical evidence that this perception extends to end-users of these technologies. The most statistically robust research [43] indicated that to the contrary, users of AR guidance systems in a heritage space receive greater reinforcement of knowledge than users of traditional audio systems or those without any guidance system. Furthermore, and according to these authors' findings, such new technologies can also improve (over traditional systems) the "sense of place" in all dimensions analyzed (place attachment, place dependence, and place identity). As a negative counterpart, it may be mentioned that users were "susceptible to distractions from external objects and engaged in fewer discussions and interactions about the historical sites with their companions. Consequently, the system tested in this study limited interpersonal interactions during visits to historical and heritage sites" [43].

On the contrary, for authors who have dealt with these aspects from a theoretical perspective, the danger in AR and VR is not merely in breaking the geographical connections established by an element of heritage, but also possibly the temporal connections. As Virilio claimed two decades ago, "the problem of Virtual Reality is, essentially, to deny the *hic et nunc*, to deny the 'here' for the benefit of the 'now'" [18] (p. 47). In this overlapping of time permitted by AR and VR, the nostalgic component linked to the idea of heritage would be lost [20], given that it would cease to trigger the temporal caesura that establishes the difference between our present and the "past other," from which those elements classified as patrimonial are received. According to Hartog, the use of AR and VR would constitute a clear case of "presentism." It would not recover history so much as employ technology to recover an emotional past to make it present and palpable [24]. In reality, and against the nostalgic feelings linked to heritage, which seek to allow us to travel in time even as we travel in space, AR and VR do not try to move visitors to the historical past, but to the emotional past of which Hartog speaks, within the visitor's own time. Rather than looking back in time, this would transport the heritage into present, breaking the caesura between past and present perceived by Choay [21].

This possible modification of temporal relationships has not represented a significant concern for most of the selected empirical studies. But this does not mean that some do not share such

presuppositions a priori, in assuming the possibility of new "connections to history and the future, thereby giving expression to concepts that have been widely discussed in the literature such as identity and belonging, diversity and intercultural dialogue, popular beliefs, and traditions" [59].

But if AR and VR acquire the capacity to generate a new "regime of historicity," they can also subvert the consideration of a heritage element that is valued because it belongs to a certain time in the past. Temporal distance, and its functional sanctification, is a quality inherent in our understanding of heritage, and this can be lost by way of technological interaction and the permanent manipulation it allows. J. Urry [32] suggested several decades ago that this had already happened in postmodern museums as, although they had been historically "premised upon the aura of the authentic historical artifact," emphasis was at that time "being placed on the participation by visitors in the exhibits themselves" [32] (pp. 118–119).

This loss of the aura of which Urry speaks can lead to certain reflections on this subject, of the kind made by philosopher W. Benjamin, in The Work of Art in the Age of Mechanical Reproduction, where he claims that the aura is a "unique phenomenon of a distance, however close it may be" [67] (p. 1). If, according to Benjamin, technical reproducibility had already destroyed the aura of the work of art by the 1930s, it seems obvious that AR and VR will definitely damage, in the case of heritage, the pre-existing relationship between the categories of space and time. In the virtual space, the past is no longer "interwoven" with the present, but instead superimposes the past origins of a given heritage over the present of those who visit that heritage, thus closing distance from the historical point of view.

Along this line follow conclusions extracted from examples analyzed by Guttentag [3], who claimed that the perception of authenticity of experiences of AR is not absolute but situated in a continuous complex between two pairs of opposites (authentic and inauthentic absolutes), in which other, older phenomena (such as historical recreations) should be included. One conclusion of this analysis is that the authenticity (and, therefore, the "aura") granted to the represented patrimonial event or object depends on a multitude of variables that may range from the personal characteristics of the user (age, gender, interest in technology, etc.) and the quality of recreations, to such aspects as the prestige of the institutions that participate in and promote such recreation [3].

In terms of the research in this study, one essential aspect of our approach is the importance that users continue to assign to the location of the consumption. Being closer to the sensuous experience of space awards greater value, and experiences undertaken far from the patrimonial space represent less value than "in-situ" experiences [3]. This seems to indicate that even the most facile users of such technology may still assign importance to the time–space frames that build the concept of heritage, and that, to some extent, they perceive as contradictory the timelessness and extra-spatiality that these technologies appear to admit. In this sense, a certain distinction can be established between the greater real/physical and symbolic separation that the space and the represented heritage elements allow by way of VR, as opposed to AR or even Mixed Reality, in which the user must be physically present in the observed place.

More unquestionably problematic is these technologies' enormous capacity to generate symbolic worlds, which theoretically entails the risk of accentuating the processes of heritage appropriation by way of selection and, later, inclusion in a determined system of representation. This is a risk already associated with the so-called heritage industry and the explosion of cultural urban tourism. The commercialization of virtual heritage realities—alien to the corporeality and sociability that the urban space implies—can accentuate the problem. This may be seen as reinforcing the emergence of "dissonant heritages," according to the ideas of Tunbridge and Ashworth [30], further promoting the alienation that many societies experience in the face of a heritage they no longer perceive as their own.

## 5. Conclusions

Thematic analysis has revealed how numerous authors consider AR and VR to be technological tools capable of facilitating new forms of access to information and diverse ways of learning, such as techniques of M-Learning and, especially, gamification. Moreover, the application of these tools to

policies for the promotion and marketing of diffuse destinations make them excellent instruments for improving visitor experience and attracting new tourist profiles. From another point of view, their technological capacities convert AR and VR into mechanisms for the protection and conservation of cultural heritage, either through digital recovery of lost cultural assets or by recreating virtual worlds that are technically inaccessible for diverse reasons.

Regarding the negative and ambivalent perspectives sometimes expressed, authors have noted that access to information generated via AR and VR can foster a temporal and spatial disconnection from the urban cultural heritage, creating virtual worlds that are completely separate from urban and/or touristic objectivity. Some prioritize the banalization of the tourist experience based on mere enjoyment of a digitalized reality, scrubbed clean of the social, economic and cultural problems of tourist destinations. In this sense, gamification, understood as a tool for transmitting information, is seen as risky by specialists when designed as expressly playful, eliding the realities of cultural heritage and the cities where it is found.

In conclusion, we have answered the questions and objectives set at out the beginning of this investigation. According to the numerous authors studied, the use of AR and VR tools can generate both positive and negative effects in the processes of interpretation of urban cultural heritage. As mentioned earlier, these are technological tools that facilitate the dissemination of information, providing a quick, easy, and novel approach to knowledge that is furthermore highly valued by the user. These instruments are equally appreciated by managers of tourist destinations, widening the exposure of territories and reinforcing the identity of the local population. However, the use of such tools entails risks related to the banalization of the visitor's experience as well as disconnection from urban reality and cultural heritage itself, in spatiotemporal terms. Although these tools remain little used by tourists, especially compared with traditional audio guides or self-guided visits, they are becoming common, which makes it necessary to deepen our understanding of their effects on users, cultural heritage, and urban tourist destinations.

The collection of new experiences related to AR and VR technologies applied to the urban cultural heritage will allow determination of the behavior of the phenomenon over the medium and long term. With the aim of outlining some of these future lines, a non-systematic analysis of the literature published in 2018 and 2019 has been carried out, based on the selection criteria discussed in the methodological section of this article [68–73]. In general, publications value AR and VR technologies as instruments not only to attract and retain visitors in tourist destinations, but also to integrate them into the development of smart destinations, as pointed out by Marasco et al. [69].

On the other hand, within this idea, articles that analyze the development of tourism applications based on AR and VR are many, including those by dela Cruz et al., by Jones et al., by Morganti and Bartolomei, and by Panou et al. [70–73]. All of these are designed to give value to historical, artistic, social, and other characteristics of urban cultural heritage, including both existing assets and those that have been lost [73]. These sites have been analyzed by users to evaluate their functionality, and assessments have been very positive in all cases. In fact, many opinions coincide with certain results obtained in this investigation: in general, they consider these types of applications as allowing the potential tourist to learn about the cultural heritage in a more playful, autonomous way that is adapted to users' personal interests and motivations [71]. In particular, users appreciate the dynamism of GPS, which allows the acquisition of knowledge *in situ*, also revealing theretofore unknown cultural elements located outside the usual tourist circuits [71,73].

In addition to these issues, others such as sustainability or sustainable development have not been detected in the publications analyzed, although these will doubtless acquire greater relevance in coming years. Future research will allow us to incorporate more publications into our study, expanding the sample to include new topics and thus allowing us to enrich our conclusions and delve more deeply into these phenomena.

**Supplementary Materials:** The following are available online at: https://docs.google.com/forms/u/1/d/1S5_m1nLy7bNTqzBlYnVzEnND-RjEevXb8FZHBBbpjzQ/edit?usp=drive_web.

**Author Contributions:** Both authors participated equally in reviewing literature connected with the topic data, as well as in analyzing data from statistics sources. Both authors participated in writing the introduction and conclusion remarks.

**Funding:** This research forms part of two competitive projects: "Amón-RA. Implementation of Virtual Reality as a tool for the enhancement and dissemination of the historic landscape of the Amón neighborhood" (CF. 1412003), a research project advanced by the School of Architecture and Urbanism and the School of Computer Engineering of the Technological Institute of Costa Rica (TEC), in collaboration with the Autonomous University of Madrid, Spain; and "Culture and Territory in Spain. Processes and impacts in small and medium-sized cities," (CSO2017-83603-C2-2-R), financed by the State Research Program "Development and Innovation Oriented to the Challenges of Society" of the Spanish Ministry of Economy, Industry, and Competitiveness, within the framework of the State Plan for Scientific and Technical Research and Innovation, 2013–2016. The project was developed by the Research Group in Urban Studies and Tourism (URByTUR) of the Department of Geography of the Autonomous University of Madrid (UAM).

**Acknowledgments:** The authors would like to thank the anonymous reviewers at Sustainability for their valuable contributions and suggestions, which have made possible significant improvements to the text.

**Conflicts of Interest:** The authors declare no conflicts of interest.

## Appendix A

| Selected references used in the Atlas.ti study |
| --- |
| • Aggoue, H. Virtual reality: Towards preserving Alexandria heritage by raising the awareness of the locals International. Journal of Architectural Research 2017, 11(3), 94. |
| • Albourae, A.T.; Armenakis C.; Kyan, M. Architectural Heritage Visualizations Using Interactive Technologies. In The International Archives of the Photogrammetry, Remote Sensing and Spatial Information Sciences. 26th International CIPA Symposium Volume XLII2//W5, 2017, 7–13. |
| • Cartwright, W. Visualizing alternative futures. In Landscape Analysis and Visulaisation. Christopher Pettit Eds.; Springer: Berlín, Germany, 2008, pp. 489–507. |
| • Chang, Y-L.; Tse Hou, H.; Pan, C. Y; Sung, Y. T; Chang, K. E. Apply an Augmented Reality in a Mobile Guidance to Increase Sense of Place for Heritage Places. Educational Technology & Society 2015, 18(2), 166–178. |
| • Fang, L. The Interdisciplinary Research of Virtual Recovery and Simulation of Heritage Buildings. Take Lingzhao Xuan in the Palace Museum as an Example. Conservation Science in Cultural Heritage 2014, 14(2), 189–205. |
| • Garau, C. From Territory to Smartphone: Smart Fruition of Cultural Heritage for Dynamic Tourism Development. Planning Practice and Research 2014a, 29(3), 238–255. |
| • Garau, C. Smart Paths for Advanced Management of Cultural Heritage. Regional Studies, Regional Science 2014b, 1(1), 286–293 |
| • Garau, C.; Ilardi, E. The 'Non-places' Meet the 'Places:' Virtual Tours on Smartphones for the Enhancement of Cultural Heritage. Journal of Urban Technology 2014, 21(1), 79–91. |
| • Guttentag, D. A. Virtual Reality: Applications and Implications for Tourism. Tourism Management. 2010, 31(5), 637–651. |
| • Hain, V.; Löffler, R.; Zajícek, V. Interdisciplinary Cooperation in the Virtual Presentation of Industrial Heritage Development. In Procedia Engineering. World Multidisciplinary Civil Engineering-Architecture-Urban Planning Symposium 2016 WMCAUS 2016 161, 2016, 2030–2035. |
| • Hincapie, M.; Diaz, C.; Zapata, M.; Mesias, C. Methodological Framework for the Design and Development of Applications for Reactivation of Cultural Heritage: Case Study Cisneros Marketplace at Medellin, Colombia. Journal on Computing and Cultural Heritage (JOCCH) 2016, 9(2), 1–8. |
| • Joo-Nagata, J.; Martínez Abad, F., García-Bermejo J.; Francisco, J.; García-Peñalvo; F. Augmented Reality and Pedestrian Navigation through its Implementation in M-Learning and E-Learning: Evaluation of an Educational Program in Chile. Computers and Education 2017, 111, 1–17. |

| Selected references used in the Atlas.ti study |
| --- |

- Kang, J. AR Teleport: Digital Reconstruction of Historical and Cultural-Heritage Sites for Mobile Phones Via Movement-Based Interactions. Wireless Personal Communications 2012, 70(4), 1443–1462.
- Koutsoudis, A.; Arnaoutoglou, F.; Chamzas, C. On 3D Reconsutruction of the Old City of Xanthi. A Minimum Budget Approach to Virtual Touring based on Photogrammetry". Journal of Cultural Heritage 2006, 8(26), 26–31.
- Malegiannaki, I.; Daradoumis, T. Analyzing the Educational Desing, Use and Effect of Spatial Games for Cultural Heritage: A Literature Review" Computers & Education 2017, 108, 1–10.
- Martínez, J. L.; Álvarez, S.; Finat, J.; Delgado, F. J.; Finat, J. Augmented Reality to Preserve Hidden Vestiges in Historical Cities. A Case Study." In The International Archives of the Photogrammetry, Remote Sensing and Spatial Information Sciences. 3D Virtual Reconstruction and Visualization of Complex Architectures Volume XL5//W4, 2015, 61–67.
- Marques, L. F.; Tenedório, J. A.; Burns, M.; Romao, T.; Birra, F.; Marques, J.; Pires, A. Cultural Heritage 3D Modelling and Visualisation within and Augmented Reality Environment, Based on Geographic Information Technologies and Mobile Platforms. In ACE: Architecture, City and Environment = Arquitectura, Ciudad y Entorno, 2017, 11(33), 117–136.
- Mendoza, R.; Badiris, S.; Fabregat, R. Framework to Heritage Education Using Emerging Technologies. In Procedia Computer Science. International Conference on Virtual and Augmented Reality in Education, 2015, 239–249.
- Menghi, R; Maino, G.; Panebarco. M. Virtual Reality Model for the Preservation of the Unesco Historical and Artistical Heritage. International Conference on Image Analysis and Processing 2011, Volume Part II, 475–485.
- Minucciani, V.; Garnero, G. Geomatics and Virtual Tourism. Proceedings of the 10th Conference of the Italian society of Agricultural Engineering 2013, 44 (2), 504–509.
- Mortara, M.; Catalano, C. E.; Bellotti, F.; Fiucci, G.; Houry-Panchetti, M.; Panagiotis, P. 2014. Learning Cultural Heritage by Serious Games. Journal of Cultural Heritage 2013, 15(3), 318–325.
- Olesksy, T.; Wnuk, A. Augmented Places: An Impact of Embodied Historical Experience on Attitudes Towards Places. Computers in Human Behavior 2016, 57, 11–16.
- Pantano, E.; Corvello, V. Tourists' Acceptance of Advanced Technology-based Innovations for Promoting Arts and Culture. International Journal of Technology 2014, 64(1), 3–16.
- Petrucco, Corrado, and Daniele Agostini. Teaching Cultural Heritage Using Mobile Augmented Reality. Journal of E-Learning and Knowledge Society 2016, 12(3), 115–128.
- Pietroni, E. An Augmented Experience in Cultural Heritage through Mobile Devices: 'Matera Tales of a City' Project. In 18th International Conference on Virtual System and Multimedia (VSMM), Milan, Italia, 2012.
- Ramsey, E. Virtual Wolverhampton: Recreating the Historic City in Virtual Reality." Archnet-IJAR: International Journal of Architectural Research 2017, 11(3), November, 42–57.
- Polat, M; Karas, I. R; Kahraman, I; Alizadehashrafi, B. AR Based App for Tourist Attraction in Eski Carsi (Safranbolu). The International Archives of the Photogrammetry, Remote Sensing and Spatial Information Sciences. In 3rd International GeoAdvances Workshop Volume XLII2/W1: 177–181, 2016.
- Rodríguez, E.; Martín-Gutierrez, J.; Meneses, M. D.; Armas, E. Interactive Tourist Guide: Connecting Web 2.0., Augmented Reality and QR Codes. In Procedia Computer Science. International Conference on Virtual and Augmented Reality in Education, 2013, 25, 338–344.
- Soliman, M. Virtual Reality and the Islamic Water System in Cairo: Challenges and Methods. Archnet-IJAR: International Journal of Architectural Research. 2017, 11 (3) November, 78–93.
- Tom Dieck, M. C.; Jung, T. A Theoretical Model of Mobile Augmented Reality Acceptance in Urban Heritage Tourism. Current Issues in Tourism, 2018, 21(2), 154–174.

**Appendix B**

**Table A1.** Description of the codes.

| Code | Description of Code. The Use of AR/VR Technology Tools |
| --- | --- |
| Digital projection | Increases the projection of urban cultural heritage in digital environments. |
| Economic cost | Incorporated into the economic management of urban cultural heritage in all its dimensions. |
| Gamification | Favors the learning of urban cultural heritage through the incorporation of gamification tools. |
| Identity | Strengthens the relationship between the user of technologies and the urban cultural heritage. |
| Improvement of accessibility | Favors accessibility for people with disabilities, or knowledge of a cultural heritage that has disappeared or is difficult to access. |
| Improvement of awareness | Reinforces the awareness of users toward the urban cultural heritage. |
| Improvement of conservation | Participates in the improvement of preservation of the urban cultural heritage. |
| Improvement of information | Improves information about the urban cultural heritage received by users of these technologies. |
| Improvement of knowledge | Improves user knowledge of cultural heritage elements located in urban environments. |
| Improvement of management and planning | Increases management, planning, and urban development that incorporates cultural heritage. |
| Improvement of preservation of historical values | Consolidates historical, artistic, cultural, and other values of urban cultural heritage. |
| Improvement of protection | Reinforces the work of protecting the urban cultural heritage and reinforcing its conservation. |
| Improvement of public–private collaboration | Encourages collaboration between public and private agents of a city working in cultural and tourism development. |
| Improvement of tourist destinations | Reinforces tourist destinations, putting value into urban cultural heritage elements. |
| Improvement of visitor experience | Improves the user experience of these technologies during visits to urban cultural heritage sites. |
| Intangible heritage | Favors the protection, conservation, and diffusion of intangible urban cultural heritage. |
| Interaction with cultural heritage | Strengthens the connection of AR/VR users with the urban cultural heritage. |
| M-Learning | Increases the awareness of mobile device users of urban cultural heritage. |
| Nomadic museography | Contributes to the development of nomadic museography by taking cultural heritage out of museums. |
| Prevention of damages | Prevents damage, disappearance, or destruction of urban cultural heritage. |
| Reactivation of the heritage | Contributes to the activation of an urban cultural heritage that may be inactive from a social, cultural, or other point of view. |
| Recovery of the vanished heritage | Favors the digital recovery of vanished cultural heritage and, therefore, contributes to disseminating knowledge about same. |
| Sensitization | Raises awareness of users to knowledge, protection, conservation, and dissemination of urban cultural heritage. |
| Smart city | Includes urban cultural heritage in new trends around the intelligent management of cities. |
| Territory management | Increases the management and development of a specific territory using economic, social, cultural, and other resources. |
| Theory of digital restoration | Contributes to the debate on new theories of digital restoration as applied to cultural heritage. |
| Virtual museum | Generates virtual museums in which the urban cultural heritage plays a major role. |
| Virtual worlds and symbolic worlds | Sets up symbolic and virtual worlds using the cultural heritage of cities as a basic resource. |

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
