# Peer review of "The Historic City, Its Transmission and Perception via Augmented Reality and Virtual Reality and the Use of the Past as a Resource for the Present: A New Era for Urban Cultural Heritage and Tourism?"

_sustainability, doi:10.3390/su11102835_

Round 1

Reviewer 1 Report

This paper could be interesting for the SUSTAINABILITY readers, but I think a long way off from being fit for publication. The article does not follow the format of the Sustainability journal (not only for references but also for the organization of the paper). I also strongly recommend to further investigate the literature, adding most  updated references and to amplify all sections.

For instance, I suggest:

Han, D. I. D., Weber, J., Bastiaansen, M., Mitas, O., & Lub, X. (2019). Virtual and augmented reality technologies to enhance the visitor experience in cultural tourism. In Augmented Reality and Virtual Reality (pp. 113-128). Springer, Cham.

Trinchini, L., & Spyriadis, T. (2019). Towards Smart Creative Tourism. In Smart Tourism as a Driver for Culture and Sustainability (pp. 451-465). Springer, Cham.

Bellini N., Pasquinelli C. (eds) Tourism in the City. Springer, Cham, 2016.

Xiao, W., Mills, J., Guidi, G., Rodríguez-Gonzálvez, P., Barsanti, S. G., & González-Aguilera, D. (2018). Geoinformatics for the conservation and promotion of cultural heritage in support of the UN Sustainable Development Goals. ISPRS journal of photogrammetry and remote sensing142, 389-406.

The title and also the abstract must be revised, because they are too generic. The abstract does not focus well on the originality of the research. 

The introduction should be structured better, the authors must focus on the state of the art of research and must indicate how they intend to develop the article. Quotes are not up to date, which questions the originality of the article.

I strongly recommend to organise better all sections: Introduction, Methodology, Results, Discussions and Conclusions (the introduction is dispersive, it should be without subparagraphs and without footnotes).

I think that in this form the paper is not suitable for publication.

Author Response

LETTER TO REVIEWERS

First of all, we want to thank the reviewers for their suggestions, which have without a doubt served to improve our original text significantly. A note of thanks has been added to acknowledge this contribution at the end of the text (lines 1169-1170). If the article is published, these acknowledgments may be introduced elsewhere, in accordance with the standards of the journal.

Also, before proceeding, we here offer an apology, both to the journal and to the reviewers, for the erroneous list of references included in our first submission (we included an old version of the bibliography which neither met the standards of the publication nor included all references used).

Now, to highlight the changes made:

REVIEWER 1

In paragraph 1, reviewer 1 states:“the article does not follow the format of the Sustainability journal (not only for references but also for the organization of the paper).”

From the formal and organizational point of view, reviewer 1 points out the need for the text to follow the recommended manner of organization. The current structure follows those recommendations (introduction, methodology and data, results and discussion, and conclusions). The only break in this structure appears in section 2 (background), which we believe is essential for understanding the text; also, the results and discussion sections are grouped under a single heading.

Note that, in addition to following the recommendations of the reviewer and the requirements of the journal, we have discovered other already published texts that, while following the recommended structure, introduce minimal changes essential to the discursive logic of the text. We believe that, at this time, and following the instructions of the reviewer, our text may fall into this category. 

In the first and penultimate paragraphs appears the following comment:“I also strongly recommend to further investigate the literature, adding most  updated references and to amplify all section”; and “The introduction should be structured better, the authors must focus on the state of the art of research and must indicate how they intend to develop the article. Quotes are not up to date, which questions the originality of the article.” The reviewer also includes a list of updated references of great interest to our work, which we greatly appreciate.

Both reviewers indicate a need for major changes to the introduction. In general (referring to the introduction and the text as a whole), it is requested that we introduce a bibliography more specialized in AR and in VR, given the need to approximate the state of the art and to expand the number of references. In fact, reviewer 2 says that the results and discussion section points to aspects related to AR/VR that have not been previously addressed.

All these recommendations have tried to take into account in the new introduction, which includes a very significant number of new references (updated, as noted by reviewer 1) on the relationship between heritage and the new tools being studied, the possibilities of these tools, their potential problems, etc. In fact the current version is about double the length of the revised text.

The introduction has also been reworked from a formal point of view, mainly in relation to sub-plots, footnotes, etc. The objectives and methods of this research have also been explained more clearly; and the hypothesis that supports our investigation has been expressly included in the last paragraphs.

In terms of organization and content, the reviewer also points out that: The title and also the abstract must be revised, because they are too generic. The abstract does not focus well on the originality of the research. 

Both aspects have been completely reworked. We believe that now they better reflect the contents, characteristics, and original elements of the text. The abstract now follows more faithfully the requirements of the publication.

Reviewer 2 Report

The paper is overall well-written (although it should be proof-read to iron our remaining grammatical errors).

However, at this stage there are a few issues that should be addressed before it can be recommended for publication.

The introduction points out key considerations derived from the literature of AR/VR in urban heritage and some interesting aspects are mentioned, such as the technology's limitation to visualize, but lacking the ability to create meaning, which dictates the importance of data input and design and implementation of technology to the context. However, the present study largely fails to consider the complete spectrum of knowledge that has so far been developed in this field. Although not all studies might be relevant for the chosen methodology, it is important to make an attempt to recognize the wide array of knowledge that has been generated in this field.

In 2.1 an interesting argument is given that questions the impact of AR/VR on authenticity of the urban heritage context. What happens with authenticity (or with perceived authenticity) of the site when virtual remodeling of artifacts and objects are perceived to be authentic? How authentic does the place appear to be, and how authentic is the constructed visualization given that it is often a selection of what was considered worthy of preservation? Leaning on the perspective brought forth by Siberman (2004), this would be expected to be discussed more critically.

Literature on AR/VR in general and in application to the urban heritage context is completely missing and should be included. The latter part of the paper discusses many topics that have already been discussed multiple times in AR/VR literature and add limited insights as a result. Instead, these points should be expanded and related to the current technological capabilities.

It remains unclear to what extent "the opinion of experts" was considered in this study. It seems to be largely based on literature analysis.

The methodology is overall well-described, but has two issues that need to be improved or justified. The current selection of 30 publications feeding into this study is defended with the outlet of publication. While this can certainly be argued from the point of expected impact, it neglects to provide a complete picture of the knowledge in this field. Only in the last 5 years, there are close to 8,000 studies related to 'augmented reality virtual reality urban heritage' in Google Scholar. 30 does not seem to be an appropriate representation of themes that were developed in the field. It would also be beneficial to state which journals the selection were published in. The considered articles in this study need to be updated to include studies published in 2018 and part of 2019. From 2018 there are over 2,000 articles published in this field.

The methodology proposes the use of questionnaires as the first step of the analysis. Based on the intention of the questionnaire, it seems to be a content analysis or narrative review. Please verify.

Some of the codes that were formulated seem rather questionable and should be reviewed to be clear and representative, e.g. 'digital projection' or 'improvement of tourist destinations' seem to be rather broad and therefore overlap with some of the other codes.

The paper focuses largely on the information that was gathered and analysed. It would be meaningful to build on these key themes and discussion that was provided to further elaborate on the expected impact of the findings on the future of urban heritage sites and historic cities as well as on the issue of sustainability and sustainable development.

Author Response

LETTER TO REVIEWERS

First of all, we want to thank the reviewers for their suggestions, which have without a doubt served to improve our original text significantly. A note of thanks has been added to acknowledge this contribution at the end of the text (lines 1169-1170). If the article is published, these acknowledgments may be introduced elsewhere, in accordance with the standards of the journal.

Also, before proceeding, we here offer an apology, both to the journal and to the reviewers, for the erroneous list of references included in our first submission (we included an old version of the bibliography which neither met the standards of the publication nor included all references used).

Now, to highlight the changes made:

REVIEWER 2

Paragraph 1

The introduction points out key considerations derived from the literature of AR/VR in urban heritage and some interesting aspects are mentioned, such as the technology's limitation to visualize, but lacking the ability to create meaning, which dictates the importance of data input and design and implementation of technology to the context. However, the present study largely fails to consider the complete spectrum of knowledge that has so far been developed in this field. Although not all studies might be relevant for the chosen methodology, it is important to make an attempt to recognize the wide array of knowledge that has been generated in this field.

As noted at the beginning, the two reviewers indicate a need for major changes in the introduction. In general it is requested that we introduce a bibliography more specialized in AR and VR, to better consider the state of the art and to increase the number of references (reviewer 1). In fact, it is expressly stated (by reviewer 2) that in the results and discussion section, certain aspects related to AR/VR have not been previously treated. All these recommendations have been taken into account in the new introduction, which features a very significant number of new references (updated, as noted by reviewer 1) on the relationship between heritage and the new tools being discussed, the possibilities of these tools, their potential problems, etc.

Paragraph 2

In 2.1 an interesting argument is given that questions the impact of AR/VR on authenticity of the urban heritage context. What happens with authenticity (or with perceived authenticity) of the site when virtual remodeling of artifacts and objects are perceived to be authentic? How authentic does the place appear to be, and how authentic is the constructed visualization given that it is often a selection of what was considered worthy of preservation? Leaning on the perspective brought forth by Siberman (2004), this would be expected to be discussed more critically.

Reviewer 2 indicates the need to complete section 2.1 with a discussion on the concept of authenticity, proposing that it be supported by the aforementioned text by Silberman (2004). An attempt has been made to cover this gap in the final six paragraphs of that sub-section, also incorporating another text by Silberman (2008) and some another new reference

Paragraph 3

Literature on AR/VR in general and in application to the urban heritage context is completely missing and should be included. The latter part of the paper discusses many topics that have already been discussed multiple times in AR/VR literature and add limited insights as a result. Instead, these points should be expanded and related to the current technological capabilities.

Please see the comments on paragraph 1 as pertain to remarks by reviewer 1.

Paragraph 4

It remains unclear to what extent "the opinion of experts" was considered in this study. It seems to be largely based on literature analysis.

Indeed, this article is based on analysis of the scientific literature made by experts in AR and VR as applied to cultural heritage in urban spaces. To make this clear, the following phrase has been reformulated on page 10:

To this end, prior academic discussion on this topic has been reviewed through the analysis of those articleswith the greatest scientific impact over the last 10 years are analyzed (2007-2017)”

Paragraph 5

The methodology is overall well-described, but has two issues that need to be improved or justified.

1.      The current selection of 30 publications feeding into this study is defended with the outlet of publication. While this can certainly be argued from the point of expected impact, it neglects to provide a complete picture of the knowledge in this field. Only in the last 5 years, there are close to 8,000 studies related to 'augmented reality virtual reality urban heritage' in Google Scholar. 30 does not seem to be an appropriate representation of themes that were developed in the field.

We understand the doubts of the reviewer regarding the representativeness of the sample used in this article, especially when compared with the vast amount of bibliographical references available via "Google Scholar". When we began to design this research, the first of our objectives was to establish a strategy to make a representative selection of the sample based on quality scientific criteria. "Google Scholar" was the first search engine we used to test our publication selection process. 

In the first place, we realized that it was impossible to analyze the number of references found through "Google Scholar" from a qualitative point of view, given the extensive volume. Secondly, in line with other authors such as Delgado, Robinson-García, Torres and Salinas (2013), who published an interesting article in the Journal of the American Society for Information Science and Technology(available at https://arxiv.org/ftp/arxiv/papers/1309/1309.2413.pdf), we value the scientific rigor of the search engine (types of publications appearing, in what position, quality levels, etc.) but find no clear answers. For this reason, we prefer to base the selection of our corpus on a review of the database of academic impact journals widely agreed upon by the scientific community: the Web of Science, managed by Clarivate Analytics, and SCOPUS, endorsed by Elsevier.

Once the scientific search engines had been chosen, we decided to apply the following criteria to the bibliographic corpus:

·     Selection of articles based on AR and VR tools applied to a cultural heritage located in urban spaces, as a consequence of our training as Human Geographers and our research specialization in the field of cities. This aspect has been expanded in the introduction. 

·     Selection of articles that, in addition to having a theoretical aspect in terms of AR and VR, provide an empirical experiment that can explain the results of applying these technologies to cultural heritage located in urban spaces. 

·     Selection of articles to be integrated over a ten-year time period (2007-2017), a chronology broad enough to obtain results and representative conclusions. The decision to close the study at 2017 is due to the fact that in that year we launched two research projects and chose to develop our research based on the 10 years prior. Even so, according to the recommendations of the reviewer (2.3), we have decided to incorporate from a non-systematized point of view (that is, not analyzed under the ATLAS.ti methodology) general conclusions on articles from 2018 and 2019. 

Taking into account the search engines used and the criteria applied in the selection of articles, the resulting sample comprises 30 publications including articles, conferences, and conference proceedings. We want to emphasize that our research is not quantitative and does not use methodologies or tools of a quantitative nature. Regarding the size of the sample in qualitative research, Salamanca and Martín-Crespo (2007), based on the criteria of Hammersley and Atkinson (2001), Gumperz (1981), and Polit and Hungler (2000), state the following: 

Regarding the size of the sample, there are no firmly established criteria or rules determined on the basis of informational needs; therefore, one of the principles that guide the sampling is the saturation of data, to the point where new information is obtained and begins to be redundant”. 

At first, we conducted the analysis of our research using only articles. Subsequently, we include conferences and conference proceedings. The results obtained through the use of more publications proved very similar, so we decided to validate the sample size at 30 publications in total, according to criteria used in these types of methodologies.

We hope the reviewer will reconsider this recommendation, since in our opinion to change these criteria might affect the characteristics and even the ultimate solidity of the research.

2.      It would also be beneficial to state which journals the selection were published in. 

At the end of the article, a bibliographic list has been included which includes the publications selected in order to apply the questionnaire, the results of which have been analyzed with the Atlas.ti program.

3.      The considered articles in this study need to be updated to include studies published in 2018 and part of 2019. From 2018 there are over 2,000 articles published in this field.

As discussed in section 2.1, some general conclusions on articles publlished in 2018 and 2019 have been included

The collection of new experiences related to AR and VR technologies applied to the urban cultural heritage will allow determination of the behavior of the phenomenon over the medium and long term. With the aim of outlining some of these future lines, a non-systematic analysis of the literature published in 2018 and 2019 has been carried out, based on the selection criteria discussed in the methodological section of this article. In general, publications value AR and VR technologies as instruments not only to attract and retain visitors in tourist destinations, but also to integrate them into the development of smart destinations, as pointed out by Marasco, Buonincontri, van Nierkerk, Orlowki and Okumus (2018) [69].

On the other hand, within this idea, articles that analyze the development of tourism applications based on AR and VR are many, including those by Cruz, Sevilla, San Gabriel, de la Cruz and Joyce (2018); by Jones, Theodosis and Lykourentzou (2019); by Morganti and Bartolomei (2018); and by Panou, Ragia, Dimelli and Mania (2018) [70] [71] [72] [73]. All of these are designed to give value to historical, artistic, social, and other characteristics of urban cultural heritage, including both existing assets and those that have been lost [73]. These sites have been analyzed by users to evaluate their functionality, and assessments have been very positive in all cases. In fact, many opinions coincide with certain results obtained in this investigation: in general, they consider these types of applications as allowing the potential tourist to learn about the cultural heritage in a more playful, autonomous way that is adapted to users’ personal interests and motivations [71]. In particular, users appreciate the dynamism of GPS, which allows the acquisition of knowledge in situ, also revealing theretofore unknown cultural elements located outside the usual tourist circuits [71] [73].”

Paragraph 6.

The methodology proposes the use of questionnaires as the first step of the analysis. Based on the intention of the questionnaire, it seems to be a content analysis or narrative review. Please verify.

Indeed, a questionnaire (check-list) was designed as the first step in the analysis of the selected publications. With this questionnaire (available at https://docs.google.com/forms/u/1/d/1S5_m1nLy7bNTqzBlYnVzEnND-RjEevXb8FZHBBbpjzQ/edit?usp=drive_web), an analysis of the content of the publications was carried out through many codes that are presented in section 3 of the article. 

To clarify the purpose of the questionnaire, the text has been modified by expanding a sentence in section 3: 

A check-list questionnaire was composed using the “Google Forms” tool with the aim of systematizing the main contents of the selected articles and conducting an analysis of their content”.

In addition, on page 10, aspects related to the functionality of the questionnaire are mentioned:

“It is necessary to clarify that the questionnaire (check-list) was not answered by any participant, but was conceived as a tool to help researchers detect and fragment information from the selected bibliography”.

Paragraph 6.

Some of the codes that were formulated seem rather questionable and should be reviewed to be clear and representative, e.g. 'digital projection' or 'improvement of tourist destinations' seem to be rather broad and therefore overlap with some of the other codes

A table has been created (Appendix I) where information on the codes is expanded, in order to improve clarity.

Paragraph 7.

The paper focuses largely on the information that was gathered and analysed. It would be meaningful to build on these key themes and discussion that was provided to further elaborate on the expected impact of the findings on the future of urban heritage sites and historic cities as well as on the issue of sustainability and sustainable development.

The information required in the last point of the article has been expanded. 

Round 2

Reviewer 1 Report

the article does not follow the format of the Sustainability journal. I strongly suggest the authors to read the Sustainability format.  The authors have greatly improved the article. The title is too long and must be changed

Reviewer 2 Report

Thank you very much for revising the manuscript based on the comments and suggestions.

It's great to see that the comments have been addressed or a justification provided. I particularly appreciate to see a much wider spectrum of literature on AR/VR in urban heritage tourism that has now been considered. Some minor adjustment is recommended, but does not have to be addressed.

The introduction is now very extensive and includes a wider spectrum on ARVR literature. Some of the paragraphs, e.g. relating to definitions of AR/VR could be moved to the 2. Background of the study to avoid the introduction to lose focus.

Thank you very much for your justification on your chosen methodology. I appreciate the added clarification in the methodology and agree with the authors that acknowledging the totality of ARVR literature in the introduction or background of the study is sufficient.

The table in the appendix is certainly helpful to clarify on the identified themes.

I wish the authors good luck in the publication process.